# Advances in Cutaneous Squamous Cell Carcinoma Management

**DOI:** 10.3390/cancers14153653

**Published:** 2022-07-27

**Authors:** Carrick Burns, Shelby Kubicki, Quoc-Bao Nguyen, Nader Aboul-Fettouh, Kelly M. Wilmas, Olivia M. Chen, Hung Quoc Doan, Sirunya Silapunt, Michael R. Migden

**Affiliations:** 1Department of Dermatology, University of Texas MD Anderson Cancer Center, Houston, TX 77030, USA; slkubicki@mdanderson.org (S.K.); quoc-bao.d.nguyen@uth.tmc.edu (Q.-B.N.); nader.aboulfettouh@uth.tmc.edu (N.A.-F.); kwilmas@mdanderson.org (K.M.W.); ochen1@mdanderson.org (O.M.C.); hqdoan@mdanderson.org (H.Q.D.); 2Department of Dermatology, University of Texas McGovern Medical School at Houston, 6655 Travis St. Suite 700, Houston, TX 77030, USA; sirunya.silapunt@uth.tmc.edu; 3Departments of Dermatology and Head and Neck Surgery, University of Texas MD Anderson Cancer Center, Houston, TX 77030, USA; mrmigden@mdanderson.org

**Keywords:** cutaneous squamous cell carcinoma, immunotherapy, chemoprevention, solid organ transplant recipients

## Abstract

**Simple Summary:**

Cutaneous squamous cell carcinoma (cSCC) is an increasingly prevalent and morbid cancer worldwide. Management of this cancer has changed significantly in the last decade through improved risk stratification and new therapies offering patients with locally advanced and metastatic disease more effective, less toxic, and more durable treatment options. Ongoing clinical trials are assessing new therapeutic options as well as optimizing existing regimens in efforts to better manage this cancer. The recent developments highlight the need for multidisciplinary care, especially for those with locally advanced and metastatic disease.

**Abstract:**

cSCC is increasing in prevalence due to increased lifespans and improvements in survival for conditions that increase the risk of cSCC. The absolute mortality of cSCC exceeds melanoma in the United States and approaches that of melanoma worldwide. This review presents significant changes in the management of cSCC, focusing on improvements in risk stratification, new treatment options, optimization of existing treatments, and prevention strategies. One major breakthrough in cSCC treatment is the advent of immune checkpoint inhibitors (ICIs) targeting programmed cell death protein 1 (PD-1) and programmed death-ligand 1 (PD-L1), which have ushered in a renaissance in the treatment of patients with locally advanced and metastatic disease. These agents have offered patients with advanced disease decreased therapeutic toxicity compared to traditional chemotherapy agents, a more durable response after discontinuation, and improved survival. cSCC is an active field of research, and this review will highlight some of the novel and more developed clinical trials that are likely to impact cSCC management in the near future.

## 1. Introduction

cSCC consists of a wide range of clinical presentations, from low-risk squamous cell carcinoma in situ (cSCCis) to high-risk, locally advanced or metastatic tumors. The risk of metastatic disease in a patient with cSCC is approximately 2–4%, most commonly to the regional lymph nodes; however, the overall risk depends on the tumor subtype, location, and patient comorbidities [1,2]. It was not until 2010 that the American Joint Committee on Cancer (AJCC) separated cSCC into an independent staging system [3]. Improved risk stratification, particularly for patients without metastatic disease, has been an active area of research culminating in staging revisions and validation of staging criteria.

Shifts in patient demographics are influencing the incidence and development of cSCC. Patients with a history of drug-induced immunosuppression, HIV, lymphoma, leukemia, certain genetic syndromes (e.g., xeroderma pigmentosum), and solid organ transplant recipients (SOTR) are at higher risk of developing cSCC with a more aggressive clinical course [4,5,6,7,8,9,10]. SOTR remains the most significant acquired risk factor, increasing the risk of developing cSCC 5- to 113-fold, as well as the risk of developing local recurrence, metastasis, and overall mortality [11]. Owing to advances in anti-rejection therapy, donor-recipient matching, antimicrobial therapy, and transplant management over the last 30 years, SOTR patients are more likely to survive transplantation and live longer; the mean survival for a patient with a renal transplant is 22.79 years [12,13]. Similarly, patients with newly diagnosed HIV on antiretroviral therapy may have lifespans similar to that of the general population [14]. The significant gains in longevity for patients with these comorbidities will further increase the incidence of cSCC.

Identifying trends in cSCC epidemiology is challenging due to a multifactorial etiology coupled with inconsistent registry tracking—currently the CDC’s National Cancer Registry does not track cSCC. The annual incidence of cSCC is estimated to be 1.8 million cases [15]. For comparison, the combined annual incidence of other cancers in the United States is estimated to be 1.9 million; however, this figure is based on cancer registries that exclude non-melanoma skin cancer (NMSC) [16]. Worldwide, there has been a rapid increase in the incidence of cSCC due, in part, to the aforementioned demographic shifts as well as increased lifespan [17,18,19,20]. Several studies suggest a more rapid rise in cSCC incidence relative to basal cell carcinoma (BCC), which is currently the most common cancer in the United States [15,18,19]. Previous studies suggested a 4:1 incidence of BCC to cSCC, but recent data suggest a ratio of 1.69:1 overall [21]. Age is a particularly important risk factor for cSCC, with a BCC to cSCC ratio of 9.63:1 for patients aged 18–39 years and 1.33:1 for patients 65 years and older [21].

Unlike BCC, cSCC typically portends a worse prognosis. It is estimated that over 15,000 people die of cSCC in the United States annually [22]. Despite melanoma having a more aggressive clinical course than cSCC, the incidence of melanoma is significantly lower, resulting in approximately half the number of deaths in the United States [23]. However, globally, melanoma results in more deaths compared to cSCC, 62,800 versus 56,100, and disability-adjusted life years, 1.7 million versus 1.2 million [24].

## 2. Determining High-Risk cSCC and Prognostic Indicators

The 2022 National Comprehensive Cancer Network (NCCN) guidelines define high-risk and very high-risk cSCC based on the presence of risk factors for local recurrence, metastasis, or death [25]. In contrast to staging systems intended to stratify patients based on risk of metastasis, the NCCN guidelines intend to direct patient management. Risk factors for the general population are listed in Table 1. In addition, total number of tumors and frequency of development are considered risk factors in high-risk groups (i.e., immunosuppression or genetic syndromes predisposing cSCC). Based on a review of older NCCN guidelines, which used a similar approach to risk stratification as the current guidelines, 87% (*n* = 231) of cSCC diagnosed at a single institution were considered high-risk based on then-current guidelines, suggesting the guidelines are highly sensitive [26]. While the 2022 NCCN guidelines contain some modifications, the high-risk category is still highly inclusive and, therefore, less useful alone at predicting risk for metastasis.

### 2.1. Staging Systems

Despite the overall low risk of metastasis, cSCC is the second most common cancer worldwide, and the absolute number of patients with metastatic cSCC is increasing. Metastatic cSCC is associated with decreased survival compared to local disease, leading to attempts to stratify patients based on risk for metastasis [1,2,27]. Though clinical and dermatoscopic features aid in recognizing cSCC, histopathologic examination is the gold standard in the diagnosis of cSCC and identifies high-risk features, including tumor thickness, differentiation, and invasion into fat and nerves. Radiologic imaging is the diagnostic modality of choice to assess for bone invasion or metastatic disease in patients with high-risk cSCC.

Several staging systems have been proposed to predict risk of metastasis in cSCC (Table 2). The 7th edition AJCC staging manual first included a cSCC-specific staging system classifying tumors by size and additional risk factors including depth, degree of differentiation, anatomic location, and invasion of nerves and underlying skeletal structures [3,28]. More recent systems developed by a group from Brigham and Women’s Hospital (BWH) and Brueninger et al., summarized in Table 2, attempted to improve upon AJCC7 [29,30]. With the release of AJCC8 In 2017, some features from BWH were included, but only consisted of staging criteria for cSCC of the head and neck [31]. The BWH and AJCC8 systems are the most commonly used staging systems in the United States.

Schmitt et al. conducted a meta-analysis comparing the performance of AJCC7 to BWH in predicting sentinel lymph node biopsy (SLNB) positivity [32]. The rate of positivity in the BWH system was 7.1% for T2a, 29.4% for T2b, and 50.0% for stage T3 compared to the AJCC7′s 11.2 %% for T2 and 60% for T4 [32]. Both systems had 0 positive cases in the T1 stage [32]. None of the cases were categorized as AJCC7 stage T3. The authors concluded that the BWH system more accurately predicted SLNB positivity.

Ruiz et al. compared AJCC8 performance with BWH when applied to a large cohort at a single center institution, focusing on metastases and deaths [33]. The high stage AJCC8 (T3/T4) and BWH (T2b/T3) patients had indistinguishable outcomes; however, the BWH high stage subset was half the size as the AJCC8, 63 versus 121 patients, respectively [33]. The BWH system had a higher specificity and positive predictive value compared to AJCC8 [33]. Venables et al. compared AJCC8 and BWH in a larger cohort from the National Disease Registration service in the UK and found similar results, noting that BWH has a higher specificity and positive predictive value compared to AJCC8, whereas AJCC8 has a slightly higher negative predictive value [34]. Overall, the data suggest that AJCC8 upstages low-risk disease, and BWH outperforms AJCC8 in identifying low-risk cSCC.

### 2.2. Sentinel Lymph Node Biopsy

Given the predilection of cSCC for regional lymph nodes, SLNB is well suited to detect occult metastatic disease. The overall rate of SLNB positivity is 13.9% and false negativity is 4.6% [35]. Patients with a positive SLN have a higher risk of recurrence, further metastasis, and death compared to SLN negative patients despite treatment with neck dissection and/or adjuvant radiation therapy (RT) [36,37]. Thus, SLNB adds prognostic value in high stage tumors but does not appear to improve patient outcomes based on the limited evidence available. The role of SLNB in management of cSCC remains unclear and further studies are ongoing.

### 2.3. Gene Expression Profiling and Tumor Biomarkers

A recently developed prognostic 40-gene expression profile test performed on biopsied cSCC tissue allows patients to be stratified into three groups based on three-year metastasis risk [38]. A retrospective analysis of 300 patients assayed determined the risk of metastasis was 9% for class 1 (low risk), 21% for 2A (high risk), and 63% for 2B (highest risk) [39]. Management recommendations have been proposed for the different risk classes in combination with the staging systems; however, gene expression profiling has not been incorporated into the NCCN or American Academy of Dermatology (AAD) guidelines [25,40].

Expression of individual tumor markers is prognostic in cSCC. Tumor expression of programmed death-1 ligand (PD-L1) carries increased risk of nodal metastasis according to a meta-analysis of seven studies (odds ratio 2.34) [41]. Additional tumor markers associated with local recurrence, perineural invasion (PNI), metastasis, and poor survival include inositol polyphosphate 5-phosphatase, p300, telomerase reverse transcriptase, CD133, thymidine kinase 1, Myb-related protein B, and long noncoding RNAs [42,43,44,45,46,47]. Further studies are needed to determine appropriate testing protocols for tumor markers in patients at risk for advanced cSCC and their implications.

### 2.4. Imaging

Staging imaging studies are not indicated for all cases of cSCC due to the low overall risk for metastasis. Imaging is most commonly employed to screen for subclinical regional metastasis. Although limited data are available regarding imaging of cSCC, it is recommended that imaging of the regional lymph node basin be considered for BWH stage T2b and T3 cSCCs [48,49,50].

In the United States, nodal basins are typically evaluated by computed tomography (CT) with contrast due to the relatively low cost, high speed, ability to assess cortical bone involvement, and spatial information provided [51]. Ultrasound (US) is another option that is 91% sensitive in detecting nodal disease in head and neck cSCC [52]. Advantages to US include avoidance of ionizing radiation and contrast, as well as potential concomitant fine needle aspiration if a concerning node is identified. However, reliable US results may depend on technician experience and facility case volume. Positron emission tomography (PET) and magnetic resonance imaging (MRI) are less frequently used for nodal basin assessment, but the latter is well suited for assessing PNI [51].

Imaging may be considered for locally advanced disease or pre-operative planning purposes [53]. Furthermore, there may be benefits to post-treatment surveillance imaging. Baseline and surveillance imaging identified nodal disease not detected on clinical exam in 21% of patients in a cohort of high-risk cSCCs [54]. There are no consensus guidelines regarding surveillance imaging, however over 75% of tumor recurrences and metastases occur within 2 years [55,56]. In addition to a clinical exam with nodal assessment, surveillance imaging with either CT or US can be considered every 4–6 months for 2 years following definitive treatment [48].

## 3. Treatment

### 3.1. Topical Treatment

Surgical excision is the gold standard treatment for cSCC, but topical therapies are often employed for lower risk cSCCis and may be utilized as an adjunct in non-surgical candidates or those who refuse surgery. The most frequently used agents are 5-fluorouracil (5-FU), imiquimod, and destructive acid (e.g., trichloroacetic acid). There are limited quality data regarding efficacy, but Bennardo et al. conducted a systematic review encompassing 49 patients treated with 5% 5-FU, 3.75–5% imiquimod, 0.1% tazarotene, and/or 80% trichloroacetic acid [57]. The response rate was 95% when topical modalities were used in combination (5-FU + imiquimod, carbon dioxide laser + 5-FU, intralesional 5-FU + topical trichloroacetic acid) and dropped to 67% for monotherapy [57]. The majority of treatment failures were in the cohort treated with tazarotene [57].

In the author’s experience, adjunctive topical therapy with either 5-FU or imiquimod is a valuable tool to decrease surgical mobidity and disfigurement, particularly for Mohs cases, in which there is residual cSCCis on a peripheral margin after clearing the invasive tumor, or when operating in a patient with field cancerization. Topical therapies are well-suited for human papillomavirus (HPV)-associated cSCCis that are multifocal (i.e., bowenoid papulosis of the genitals) and have been reported as an adjunct when surgical margins are positive in HPV-associated cSCC [58].

### 3.2. Destruction

Destruction with liquid nitrogen or curettage and electrodesiccation (C&E) are treatment options for low-risk cSCC. A pooled analysis of eight retrospective observational studies revealed a recurrence rate of 1.7% and 0.8% for C&E and cryotherapy, respectively [59]. However, most of the lesions in this analysis were smaller, lower risk cSCC. The recurrence rate of cSCC greater than 2 cm treated with C&E was 11.8% [59]. A disadvantage of destruction, especially on the head and neck, is cosmesis. Seventeen percent of patients were deemed to have a “poor” cosmetic outcome in one study, and 54% had a satisfactory cosmetic outcome [59].

### 3.3. Surgical Excision

Surgical excision is effective for most cSCCs. Both NCCN and AAD recommend 4- to 6-mm clinical margins for standard excision of low-risk cSCC, with the AAD specifically recommending excision to the depth of the subcutaneous adipose tissue [25,40].

### 3.4. Mohs Micrographic Surgery

Mohs micrographic surgery (MMS) results in superior histopathologic verification of complete tumor extirpation and allows for maximum conservation of tissue compared to standard surgical excision [59]. This technique is especially important for cSCCs located in high-risk anatomic sites, including those on the head and neck. Rowe et al. conducted a systematic review, demonstrating a 5-year local recurrence rate of 3.1% for primary cSCC treated with MMS [27]. For recurrent cSCC, the 5-year recurrence rate after MMS was 10.0% versus 23.3% for standard excision [27]. The findings are summarized in Table 3. No RCTs or prospective cohort studies comparing MMS to other treatment modalities for cSCC have been published.

### 3.5. Radiation Therapy

RT is an option for primary treatment of cSCC when surgery is contraindicated, associated with high morbidity, or not preferred by the patient. Because of potential long-term sequelae, patients over the age of 60 years old are preferred candidates. High-level evidence of this treatment modality is lacking; current data are limited by small patient numbers, variable follow-up durations, and heterogeneity of RT modality employed. Cure rates of RT have been shown to be lower than standard surgical excision [59]. Smaller and thinner tumors may be more responsive to radiation therapy. Other disadvantages of RT include lack of margin control, prolonged course of therapy, post-radiation skin changes (chronic dermatitis and fibrosis), and potential increased risk of future cancers in the treatment field. RT as an adjuvant modality is often employed following surgical treatment of high-risk SCC and will be discussed later [60,61].

### 3.6. Photodynamic Therapy

Photodynamic therapy (PDT) is widely used for field treatment actinic keratoses and involves topical application of a photosensitizer, such as aminolevulinic acid or methyl aminolevulinate. Due to reports of high recurrence with PDT, as well as a lack of high-level evidence for its use, PDT is generally not recommended for cSCC [59,62,63]. PDT is better suited for cSCCis, which is confined to the epidermis and within the therapeutic range of PDT. A PDT cure rate for cSCCis of 86–93% can be achieved, with a 2-year sustained clearance rate of 68–71% [62].

### 3.7. Laser

Limited data exist for laser treatment of cSCC in the literature [59]. No study has shown that laser has efficacy in treating cSCC; however, case series do outline successful treatment of cSCCis with ablative laser therapy [64]. In the largest case series, 44 patients with 48 cSCCis treated with one or more passes of carbon dioxide laser at 2 W/cm^2^ demonstrated a 6.8% recurrence rate at 18 months mean follow-up [65].

Laser-assisted drug delivery augments the distribution and penetration of topically applied treatments by utilizing ablative fractional laser (AFL). Two randomized controlled trials showed clearance of cSCCis with AFL-PDT was 87.0–87.5% versus 50.0–55.3% for the PDT-only arm at 12 months [66,67]. A case series of 16 patients demonstrated 100% clearance of cSCCis at 8 weeks after a single treatment of AFL followed by topical 5% 5-FU under occlusion for 7 days [68].

### 3.8. Cytotoxic Chemotherapy

Locally advanced or metastatic cSCCs not amenable to surgical excision or radiation therapy require a more aggressive therapeutic approach [69]. Prior to the advent of immunotherapy, cytotoxic chemotherapies such as carboplatin, cisplatin, bleomycin, and 5-FU were the mainstay of advanced cSCC treatment, either as a monotherapy or combination therapy [69,70,71]. Most clinical data supporting these treatment regimens are limited single-arm studies and case series with response rates ranging from 17% to 84% [69,72,73,74]. Combination therapies demonstrate higher efficacy than single agent therapy; however, there is a lack of durable response following cessation of treatment [69]. In many studies, chemotherapies were combined with other treatment modalities such as RT or surgery, which further confounds long-term data analysis of these treatment regimens [75]. Cytotoxic agents demonstrate short progression free survival (PFS) and/or overall survival (OS) [74]. In cases with longer tumor remission, patients received subsequent surgery or RT [75].

Treatment-related toxicities are a significant concern for patients undergoing cytotoxic chemotherapy [69]. More than one-third of patients treated with combination therapy experienced grade 3 to 4 hematologic adverse events, including anemia and neutropenia, in available studies [71,76]. Cytotoxic agents are not FDA-approved for the treatment of advanced cSCC and are not preferred due to substantial treatment-associated morbidity and lack of durable response [70]. Due to a lack of higher-powered randomized controlled trials analyzing cytotoxic chemotherapy regimens, no formal treatment guidelines have been established.

### 3.9. Immunotherapy

Intralesional interferon represented the first use of immunotherapy for the treatment of cSCC in 1992 [77,78]. Since then, there has been an increased understanding of the immune system’s role in cSCC development, surveillance, and targeting of cancerous cells for elimination [69]. Chronic immunosuppression significantly increases the incidence of NMSC through impaired immunosurveillance.

Repeated skin exposure to mutagens such as ultraviolet radiation is a major pathogenic factor in the development of cSCC [79]. Tumor cells escape immunosurveillance by activation of regulatory checkpoints that place a physiologic brake on the immune system [80]. Immunotherapy with ICIs promotes T-cell mediated tumor destruction through the removal of the brakes on the immune system [69]. Highly mutated tumors express more tumor-associated neoantigens and are more likely to respond to ICI therapy [81,82,83]. cSCC has a relatively high number of mutations compared to other solid tumors, making it well suited for ICI therapy [69,81,82].

#### 3.9.1. Program Death 1 Inhibition

PD-1 is a transmembrane receptor expressed on T-cells, B-cells, and natural killer cells and serves as an immune regulating checkpoint. PD-L1, its ligand, is expressed on tumor and antigen-presenting cells [84]. The binding of PD-L1 to the PD-1 receptor leads to peripheral T-cell exhaustion and immunosuppression in the local tumor microenvironment [84,85]. In 2018, the introduction of immunotherapy with anti-PD-1 antibodies resulted in a paradigm shift in the treatment of advanced cSCC, offering increased efficacy, improved safety, and durability after treatment discontinuation [69].

Cemiplimab is a human monoclonal immunoglobulin G4 antibody directed against PD-1 [86]. In 2018, it became the first FDA-approved treatment for locally advanced or metastatic cSCC. It is approved in the European Union for the same indications. Approval was driven by two clinical trials involving 137 patients with advanced and/or metastatic cSCC [87,88]. In the phase I and II trials, patients received intravenous cemiplimab at a dose of 3 mg/kg every 2 weeks for up to 48 weeks (phase 1), 96 weeks (phase 2), or until there was disease progression or unacceptable toxicity [86]. The combined phase I and II populations had an objective response rate (ORR) of 47% [86,89]. Sixty-five percent of patients had durable disease control, defined as greater than 105 days without disease progression, and 54% of patients achieved disease control of at least 6 months in the phase I cohort [86]. Sixty-one percent of patients with metastatic cSCC in the Phase II study achieved durable disease control, with 57% achieving disease control for at least 6 months [86]. The majority of adverse reactions included grade 1 or 2 reactions such as fatigue, rash, nausea, diarrhea, and constipation [86,90,91]. Grade 3 or higher adverse reactions occurred less frequently, and included hypo- and hyperthyroidism, and type 1 diabetes mellitus, pneumonitis, colitis, hepatitis, nephritis, hypertension, adrenal insufficiency, and anemia [86,90,91].

A 43-month follow-up analysis of the phase II study (summarized in Table 4) showed a stable ORR of 47.2%, with improvement seen in the cohort with metastatic cSCC, which had an ORR increase from 42.9% to 46.4% [92]. Two additional patients had a complete response (CR). The improvements in ORR and CR demonstrate the findings of greater response with longer follow-up. Duration of response continued to improve across all cohorts, and the safety profile remained similar to prior studies with anti PD-1 agents [86,90,91].

Pembrolizumab is a humanized monoclonal antibody that binds the PD-1 receptor on T-cells that received FDA approval in 2020 for the treatment of recurrent or metastatic cSCC not amenable to surgery or RT. Approval stemmed from the KEYNOTE-629 Phase II clinical trial of 105 patients with recurrent or metastatic cSCC with a median follow-up of 11.4 months [93]. Eighty-six percent of patients had previously been treated with at least one systemic therapy prior to the trial. Pembrolizumab was administered at a dose of 200 mg every 3 weeks for a maximum of 24 months or until disease progression or toxicity. The combined recurrent and metastatic cSCC cohort had an ORR of 34.3%. Most adverse reactions were grade 1 or 2 and included diarrhea, fatigue, pruritus, and constipation. Grade 3 or higher adverse reactions occurred frequently and included colitis, hepatitis, endocrinopathies, pneumonitis, nephritis, and skin toxicity.

Hughes et al. subsequently analyzed the KEYNOTE-629 trial (summarized in Table 5) at a mean follow-up of 27.2 months and a new cohort of 54 patients with locally advanced cSCC treated with pembrolizumab at a mean follow-up of 14.9 months [94]. The ORR increased slightly to 35.2% in the recurrent and metastatic cohort, and the locally advanced cohort had an ORR of 50.0%. Of note, efficacy comparisons across trials are difficult to interpret, particularly because locally advanced cSCC in the KEYNOTE-629 clinical trial did not include recurrent disease, while locally advanced cSCC in the cemiplimab trials included recurrent disease [93,94]. Eleven percent of patients experienced grade 3 to 5 treatment related adverse reactions, and 8% of patients had grade 3 to 5 immune related adverse reactions [94]. Overall, a durable response and robust anti-tumor response were observed in both cohorts with an acceptable safety profile.

#### 3.9.2. Cytotoxic T-Lymphocyte-Associated Antigen 4 Inhibition

Cytotoxic T-lymphocyte-associated protein 4 (CTLA-4) is an immune checkpoint receptor expressed on T-cells. The binding of CTLA-4 to B7, its costimulatory protein receptor on antigen presenting cells, decreases T-cell regulatory activation [85,95]. Blockage of this pathway increases anti-tumor response through T-cell activation [83].

Ipilimumab is a human monoclonal antibody against CTLA-4 and is FDA-approved for treatment of unresectable or metastatic melanoma [96]. While groundbreaking for the treatment of melanoma, ipilimumab has a much smaller role for cSCC. Day et al. reported a case of a patient with both metastatic melanoma and metastatic cSCC treated with ipilimumab monotherapy for 4 cycles [97]. The patient demonstrated clinical benefit and a durable response of all tumors with grade 2 or less adverse events [97]. Larger randomized controlled trials are necessary to assess anti-CTLA-4 antibodies for cSCC treatment. Given the side effect profile of ipilimumab, other ICIs are typically favored.

### 3.10. Targeted Therapy

The epidermal growth factor receptor (EGFR) is a tyrosine kinases receptor involved in signaling pathways critical for cell proliferation, migration, differentiation, regulation, and survival [98]. Tumor development often involves abnormal activation of the EGFR pathway, and up to 80% of cSCC and 100% of metastatic cSCC overexpress EGFR [99,100]. Commercially available EGFR inhibitors include the oral small molecule tyrosine kinase inhibitors, erlotinib and gefitinib, and intravenous monoclonal antibodies, cetuximab and panitumumab [101]. Although not considered first line systemic therapy, EGFR inhibitors are considered in patients with contraindications to ICIs or ineligible for clinical trials. EGFR inhibitors may be used alone or in combination with conventional chemotherapy or RT [25].

Data supporting the efficacy of EGFR inhibitors in the treatment of cSCC are limited to a few prospective studies that are summarized in Table 6. Cetuximab monotherapy showed an ORR of 28%, including a CR in 6% and partial response (PR) in 22%, when used for at least 6 weeks in a phase II clinical study [100]. PFS and OS were 4.1 months and 8.1 months, respectively [100]. Eighty-seven percent of patients experienced a grade 1 or 2 acneiform eruption [100]. Sixty-one of patients experienced a grade 3 and 4 toxicity, including infection, bleeding of the tumor, and infusion reaction [100]. A small, 16-patient study of panitumumab monotherapy demonstrated an ORR of 31%, including CR in 13% and PR in 19% [102]. PFS and OS were 8 months and 11 months, respectively. Notably, all patients included in the study were being treated as second line therapy. Five of the sixteen patients experienced grade 3 or 4 toxicities, including four with skin toxicity [102]. Oral gefitinib demonstrated an ORR of 16%, all of which were PR, in a phase II study of 37 patients with cSCC [103]. PFS and median OS were 3.8 months and 12.9 months, respectively [103]. Oral erlotinib demonstrated an ORR of 10%, all of which were PR, in a phase II trial of 29 patients with cSCC [104]. PFS and OS were 4.7 months and 13 months, respectively [104]. Treatment with these agents may be limited by shorter duration of response and progression free survival compared to more modern systemic therapies.

Side effects of EGFR inhibitors are typically less severe than those seen with cytotoxic chemotherapy. The most common adverse event is an acneiform eruption, reported in 50–100% of patients and a major cause of drug cessation or poor treatment adherence [105]. Interestingly, the occurrence and severity of the acneiform eruption is positively correlated with tumor response [100,106]. Patients who developed this eruption secondary to EGFR inhibitors have an increased PFS and a tendency for a longer OS [100]. Diarrhea is also a common adverse effect of EGFR inhibitors due to the expression of EGFR in epithelial cells of the gastrointestinal tract, with incidences ranging from 27% to 87% in clinical trials for a variety of malignancies [107]. Other reported side effects include fatigue, rash, malaise, infection, neutropenia, peripheral sensory neuropathy, weight loss, and elevated liver transaminases [48]. Despite these adverse effects, EGFR inhibitors tend to be better tolerated by patients compared to cytotoxic chemotherapies.

### 3.11. Intratumoral Injection

cSCC is most commonly limited to the skin, allowing direct access of therapeutics to the tumor site. While systemic agents can elicit a variety of adverse events, intratumoral injection increases local drug concentrations and potentially diminishes systemic toxicities [108]. Intralesional treatment may be preferred in poor surgical candidates or those with challenging tumor location or high tumor burden. Intratumoral injections are user-dependent, and correct injection technique is crucial to the successful delivery of a drug.

#### 3.11.1. Methotrexate

Methotrexate (MTX) is an anti-metabolite therapy commonly used for its anti-tumoral and immunosuppressive effect against tumors and autoimmune disease [109]. As a competitive inhibitor of dihydrofolate reductase, it prevents the conversion of dihydrofolate to tetrahydrofolate, a step crucial to de novo purine and pyrimidine synthesis [110]. Classically, systemic MTX was used intravenously for a variety of malignancies, including childhood leukemia [111]. More recently, methotrexate has been used intralesionally in easily accessible tumors, particularly cSCC.

Intralesional methotrexate has had some success in the treatment of localized cutaneous squamous cell carcinoma without metastasis. In a single-center, prospective study that included 65 cases of cSCC treated with intralesional MTX prior to surgical treatment, 92.3% of patients showed a clinical response, with 58.5% of patients having no residual dysplasia on the subsequent excision pathology [112]. The mean tumor surface area of clinical responders was 1.69 cm^2^ versus 7.72 cm^2^ in the non-responder group, and all tumors were treated with the same protocol, two 20 mg treatments spaced 1 week apart, regardless of the tumor size [112]. Annest et al. treated 38 keratoacanthoma-type cSCC with intralesional MTX and observed a cure rate of 92% [113]. On average, two injections were given, 18 days apart, with an injection volume of 1 mL and 17.6 mg/mL of MTX [113]. MTX is metabolized in the liver and renally excreted. Although injection is intralesional, systemic adverse events have been reported [113]. MTX toxicity is characterized by nausea, vomiting, diarrhea, pancytopenia, liver dysfunction, acute renal failure, pulmonary symptoms, stomatitis, mucositis, and gastrointestinal or cutaneous ulcerations [114].

#### 3.11.2. 5-Fluorouracil

5-FU is fluoropyrimidine that is converted to its active metabolite in the cell and disrupts RNA synthesis [115]. Intralesional 5-FU has shown success in the treatment of keratoacanthomas. In 1978, Odom et al. treated 14 patients with 26 keratoacanthomas using weekly intralesional 5-FU [116]. All but one lesion cleared with an average of 2.8 injections, and of the 13 patients observed, none had recurrence at 12 months. Intralesional 5-FU has shown efficacy in patients with large keratoacanthomas, recurrent keratoacanthomas, and invasive SCC [117]. Larger-scale randomized controlled trials are needed to better characterize the use of intralesional 5-FU for the treatment of cSCC. In general, 5-FU toxicities include fever, mucositis, nausea, vomiting, diarrhea, and myelotoxicity [118].

### 3.12. Combination Therapy and Adjuncts

While surgical excision of cSCC is often the standard of care, combining therapeutic modalities has been common practice since for high-risk cSCC. Systemic therapy can be coupled with surgery or RT as adjuvant or neoadjuvant therapy [119].

Adjuvant RT reduces the likelihood of recurrence and metastasis after surgical excision [53]. However, the majority of studies on adjuvant RT have been retrospective in design [119]. A retrospective study by Wang et al. showed a higher survival rate for patients with cSCC and cervical lymph node involvement treated with surgery plus adjuvant RT, compared to those treated with surgery alone [120]. Another study by Trosman et al. showed that adjuvant RT and surgery did not significantly improve 2-year disease free survival rates compared to surgery alone [121]. These contradictions continue in numerous studies investigating adjuvant RT, and there is little evidence to help determine which high-risk cSCC tumors would benefit from adjuvant RT. AAD guidelines recommend adjuvant RT for primary cSCC with concerning PNI or otherwise high risk for regional or distant metastasis, following surgical treatment [40]. However, the guidelines do note that there is no high-level evidence about the effectiveness of this approach [40]. The NCCN recommends adjuvant RT for any cSCC with extensive perineural involvement, large nerve involvement, or in which tissue margins remain positive after definitive surgery [25].

Adjuvant chemotherapy includes the use of various antitumor agents including platinum-based agents (e.g., cisplatin and carboplatin) as well as anti-metabolites (e.g., 5-FU). Similar to adjuvant RT, studies on adjuvant chemotherapy are predominantly retrospective in nature with small patient cohorts [53]. Due to the significant toxicities of chemotherapy, these agents are not frequently used in patients with significant comorbidities [122]. Future prospective trials are necessary to optimize which locally advanced and metastatic tumors would benefit from adjuvant chemotherapy, adjuvant RT, or a combination of the two.

For locally advanced cSCC, consideration of immunotherapy and/or RT is recommended when surgery is unable to clear the tumor or would result in significant disfigurement or loss of function [25]. Numerous trials are investigating a combination of targeted therapies and immunotherapies in combination with chemotherapy or RT in an adjuvant and neoadjuvant setting [53]. At this time, due to a lack of prospective randomized controlled studies, a multidisciplinary approach and consideration of referral to a cancer center is recommended for discussion to optimally combine these treatment modalities for patients with locally advanced and/or metastatic cSCC.

## 4. Chemoprevention

Oral retinoids have been shown to reduce the incidence of cSCC in patients with certain high-risk factors—SOTR, genetic predisposition (e.g., xeroderma pigmentosum), and prior PUVA exposure; however, the retinoids do not produce a durable response and benefits quickly abate after cessation of therapy [123,124,125,126,127,128,129]. The benefit is more pronounced in SOTR patients with a prior history of cSCC, compared to SOTR patients without [125,127]. Topical retinoids and oral isotretinoin have failed to show benefits in preventing cSCC development [130,131,132]. There are conflicting data regarding the benefits of oral retinol [132,133]. Patients on oral retinoid therapy require regular monitoring given the risks, which include teratogenicity, hepatitis, hyperlipidemia, alopecia, and mucocutaneous dryness. Based on the limited data, the AAD recommends against the use of retinoids for chemoprevention, with the sole exception that acitretin may be used in patients with a history of SOTR and cSCC [40]. The NCCN acknowledges that oral retinoids may be effective in reducing the development of cSCC in some high-risk patients other than SOTR [25].

The benefits of chemoprevention are less distinct for other therapies and patient groups at high risk. Weinstock et al. demonstrated a 75% reduced risk of developing cSCC at 12 month follow-up with the prophylactic use of 5-FU cream compared to placebo in patients with a history of NMSC [134]. Study patients were included if they had a history of two keratinocyte tumors in the last 5 years, one of which was located on the head or neck. Patients with a history of SOTR, genetic predisposition, PUVA, CTCL, or prior radiation were specifically excluded from the study.

Nicotinamide, a variant of vitamin B-3, has shown efficacy in the chemoprevention of actinic keratoses and NMSC in those at elevated risk. In a phase 3 double-blinded randomized controlled trial, nicotinamide 500 mg BID resulted in a 30% reduction in cSCC after 12 months of therapy [135]. Smaller reductions were found in the incidence of basal cell carcinoma (20%) and actinic keratoses (13%). Patients were included if they had a history of 2 NMSC in the past 5 years, and those with immunosuppression or genetic skin cancer syndromes were excluded. Similar to oral retinoids, the chemopreventive benefit did not persist significantly after discontinuation of therapy. A meta-analysis of the use of nicotinamide in preventing cSCC, which included patients with a history of SOTR, found an overall rate ratio of 0.48 (0.26–0.88, 95% CI) further supporting its use as a chemopreventive agent [136]. Unlike a related drug, niacin, which often causes flushing and other vasodilatory phenomena, nicotinamide is well tolerated. There was no significant difference in adverse events between the treatment and placebo arm. The AAD guidelines recommend against the use of nicotinamide for chemoprevention; however, these guidelines were last updated in 2018 and based on the limited evidence available at the time [40]. The NCCN guidelines, most recently published in May 2022, acknowledge that nicotinamide may reduce the development of cSCC, but do not recommend for or against its use [25].

Capecitabine is an oral precursor to 5-FU, an agent commonly used as a topical treatment of NMSC and actinic keratoses. Lewis et al. first reported capecitabine-associated inflammation of actinic keratoses in a patient treated for metastatic colorectal carcinoma [137]. This and other subsequent reports led to studies evaluating the use of capecitabine as a chemopreventive agent. In a systematic review, 13 of 18 SOTR patients demonstrated at least a 50% reduction in cSCC during 12 months of therapy with capecitabine [138]. Fifty-six percent of all patients included in the review discontinued therapy due to disease progression or adverse events, including an unrelated death in one patient [138]. Interestingly, capecitabine may also reduce the incidence of basal cell carcinoma in patients with a history of SOTR [139]. Larger studies are needed to better define the chemopreventive benefits. Common adverse effects included fatigue, hand-foot syndrome, diarrhea, nausea/emesis, mucositis, anemia, and hyperuricemia/gout. Further investigation is necessary to better determine the risks and benefits of capecitabine in the prevention of cSCC; however, it may be considered in patients with very high rates of cSCC formation and field cancerization who have failed other treatment strategies.

Despite early evidence that NSAIDs may decrease the risk of cSCC, a subsequent meta-analysis showed no chemopreventive benefit [140]. Both the NCCN and AAD advise against the use of NSAIDs for chemoprevention of cSCC due to the lack of efficacy and increased risk of cardiovascular and gastrointestinal adverse events [25,40].

PDT is widely used for field treatment of actinic keratoses and has been used off-label for multiple indications, including the treatment of low-risk NMSC, as discussed earlier. Data on the use of PDT for prevention of NMSC are limited. One study of 12 SOTR patients found a 79% and 95% reduction in the number of cSCC at 12 and 24 months, respectively, when treated with cyclic PDT every 4–8 weeks [141]. Adverse events were mild and included erythema, edema, desquamation, and crusting. There were significant limitations to the study, including a short observation period and a high number of cSCCs treated at baseline immediately prior to PDT, possibly overstating the benefits of the treatment. Further investigation of this relatively low-risk preventative modality is warranted.

## 5. Future Directions

As of May 2022, there are 179 clinical trials for cSCC. The biggest trend with the current trials is the use of ICIs, including three novel agents targeting PD-1 and/or PD-L1 [142,143,144]. Many trials are investigating the efficacy of neoadjuvant, adjuvant, or combined neoadjuvant and adjuvant ICI therapy with surgery and/or RT. Several novel agents are being combined with existing ICI therapy, specifically medications stimulating IL-2, IL-7, IL-15, TLR-7/8, TLR-9, or inhibiting c5a and EGFR/TGFβ [145,146,147,148,149,150,151,152]. In an effort to decrease systemic exposure and risk of ICIs, one trial is evaluating low dose intratumoral injection of cemiplimab [153].

Investigational drugs targeting CD-47, CD-40, TGFβ1, COX-2, and the STING pathway are being studied as monotherapy for cSCC [154,155,156]. Several unique types of therapy delivery are also being studied. There are two trials evaluating photoimmunotherapy with ICIs [157,158]. Photoimmunotherapy involves systemic infusion of a medication followed by local activation with a specific wavelength of light. NCT05377905 utilizes a microneedle array that delivers intratumoral doxorubicin; notably, patients with a history of SOTR are included in this study [159]. Several studies are evaluating specific applications of RT, including the use of diffuse alpha-emitter RT, which involves intratumoral placement of radium-224 and short-range emission of alpha particles, potentially limiting collateral tissue damage [160]. IFX-Hu2.0 is a DNA plasmid that induces tumoral expression of streptococcal antigen, potentially increasing immunosurveillance, and the plasmid vehicle potentially avoids side effects associated with viral-mediated gene therapy [161,162].

Oncolytic viruses, viruses that have been genetically modified or selected for anti-tumoral activity, have been used previously in melanoma and other solid tumors. Talimogene laherparepvec (T-VEC) is a genetically modified herpes virus used in the treatment of stage III and IV melanoma and was the first to be FDA approved in the US. T-VEC is currently being studied in small trials for the treatment of cSCC as monotherapy or in combination with panitumumab or nivolumab [163,164,165]. TBio-6517, ONCR-177, RP-1, MEM-288, and Daromun are oncolytic viruses currently in phase 1 and 2 trials. They influence both tumor and tumor microenvironment with activity on Flt3, CTLA4, IL-2, IL-12, CCL4, PD-1, CM-CSF, TNF, and GALV-GP R-. TBio-6517 and RP-1 are the farthest along, with trials combining intratumoral injection of the oncolytic virus with systemic ICIs [166,167].

Treatment of locally advanced and metastatic cSCC in SOTR patients poses significant challenges. First-line treatments, including RT with chemotherapy or EGFR inhibitors, are limited by their modest efficacy and low tolerability. ICI treatment is limited by the risk of grafted organ rejection, which approaches 42% in renal transplants [168]. NCT04339062 is evaluating the use of cemiplimab in patients with a history of renal transplant, incorporating pre-infusion prednisone and sirolimus to reduce the risk of graft rejection [169]. Intratumoral RP-1 is also being investigated as monotherapy in patients with SOTR [170].

Given the variable and dynamic response of ICI therapy, several trials are assessing factors that may better predict response to therapy. Based on data from murine studies and patients with melanoma, two trials are evaluating fecal microbiota, including one that involves fecal transplant from ICI responders to ICI non-responders [171,172]. Two large cohort studies are investigating various factors and their potential association with overall survival, incidence of immune-related adverse events, and quality of life [173,174].

## 6. Conclusions

Traditional surgical and destructive methods remain the standard of care for low-risk cSCC; however, management of high-risk, locally advanced, and metastatic cSCC has evolved significantly in the last decade to include more efficacious, durable, and tolerable therapeutic options. With the increasing incidence of cSCC and growing number of patients with high risk factors, it is imperative that clinicians continue to expand their treatment armamentarium to include the most recent advances.

## Figures and Tables

**Table 1 cancers-14-03653-t001:** National Comprehensive Cancer Network features of high-risk and very high-risk cutaneous squamous cell carcinoma (May 2022); PNI, perineural invasion.

High-Risk cSCC	Very High-Risk cSCC
Tumors on the head, neck, hands, feet, pretibial, and anogenital region or trunk and extremities with diameter ≥ 2 cm	Tumors with diameter ≥ 4 cm on any location
Acantholytic, adenosquamous, or metaplastic subtypes, with PNI	Poorly differentiated, desmoplastic, >6 mm thickness, or invasion beyond fat
PNI ≥ 0.1 mm	PNI with tumor within nerve sheath of a nerve lying deeper than the dermis or ≥0.1 mm
Recurrent tumor	Lymphatic or vascular involvement
History of immunosuppression, prior site of radiotherapy, rapidly growing tumor, neurologic symptoms	

**Table 2 cancers-14-03653-t002:** Summary of four staging systems for cutaneous squamous cell carcinoma. PNI, perineural invasion.

Staging System	Stage	Risk Factors	High-Risk Factors (If Applicable)
AJCC7	T1	Tumor diameter ≤ 2 cm with <2 high-risk factors	>2 mm thickness Clark level ≥ 4 PNI Primary site ear or lip Poorly differentiated
	T2	Tumor diameter > 2 cm, or any size with ≥2 high-risk factors	
	T3	Tumor with invasion of orbit, maxilla, mandible, or temporal bone	
	T4	Tumor with invasion of other bone, or direct PNI of skull base	
AJCC8	T1	Tumor diameter < 2 cm	
	T2	Tumor diameter ≥ 2 cm and <4 cm	
	T3	Tumor diameter ≥ 4 cm, or minor bone erosion, or PNI, or deep invasion	
	T4	Tumor with gross cortical bone/marrow invasion	
BWH	T1	No high-risk factors	Tumor diameter ≥ 2 cm PNI ≥ 0.1 mm Poorly differentiated Tumor invasion beyond fat
	T2a	1 high-risk factor	
	T2b	2–3 high-risk factors	
	T3	≥4 high-risk factors, or bone invasion	
Brueninger et al.	Clinical stage (cT)	Low risk: Tumor diameter ≤ 2 cm High risk: Tumor diameter > 2 cm	
	Pathological stage (pT)	No risk: Tumor thickness ≤ 2 mm Low risk: Tumor thickness > 2 mm and ≤6 mm High risk: Tumor thickness > 6 mm	
	Co-risk factors	Immunosuppression Desmoplastic type or poor differentiation Primary site ear	

**Table 3 cancers-14-03653-t003:** Recurrence of localized cutaneous squamous cell carcinoma after surgery; cSCC, cutaneous squamous cell carcinoma.

	Primary cSCC	Recurrent cSCC
Mohs	3.1%	10.9%
Standard Excision	10.0%	23.3%

**Table 4 cancers-14-03653-t004:** Expanded follow-up of phase 2 cemiplimab data; ORR, objective response rate; PFS, progression free survival (in months).

Group	Cohort	Dosing	Patients (*n*)	ORR	Median PFS
1	Metastatic cSCC	3 mg/kg IV q2 weeks	59	50.8%	18.4
2	Locally advanced cSCC	3 mg/kg IV q2 weeks	78	44.9%	18.5
3	Metastatic cSCC	3 mg/kg IV q3 weeks	56	46.4%	21.7
Total			193	47.2%	18.5

**Table 5 cancers-14-03653-t005:** Expanded follow-up of phase 2 pembrolizumab data; ORR, objective response rate; PFS, progression free survival (in months); NR, not reached.

Cohort	Dosing	Patients (*n*)	ORR	Median PFS
Locally advanced cSCC	200 mg IV q3 weeks	54	50.0%	NR
Recurrent/metastatic cSCC	200 mg IV q3 weeks	105	35.2%	5.7
Total		159	40.3%	NR

**Table 6 cancers-14-03653-t006:** Prospective study summaries for targeted therapy of cutaneous squamous cell carcinoma; ORR, objective response rate; CR, complete response; PR, partial response; PFS, progression free survival (in months); OS, overall survival (in months). * All patients treated as second line.

Route	Therapy	ORR	CR	PR	Median PFS	Median OS
Oral	Erlotinib	10%	0%	10%	4.7	13
	Gefitinib	16%	0%	16%	3.8	12.9
Intravenous	Cetuximab	28%	6%	22%	4.1	8.1
	Panitumumab	31% *	13% *	19% *	8 *	11 *

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
