# Peer review of "Advances in Cutaneous Squamous Cell Carcinoma Management"

_cancers, 2022, doi:10.3390/cancers14153653_

Round 1

Reviewer 1 Report

The manuscript “Advances in Cutaneous Squamous Cell Carcinoma Management” is a review article regarding the actual state-of-art on recent changes in the management of cSCC, including improvements in risk stratification, new treatment options, optimization of existing treatments, and prevention strategies.

I really appreciate the work performed by authors, since the manuscript is very well written, giving a good overview on the topic. There are some concerns that authors should address in order to consider the manuscript suitable for publication:

1.      In the introduction section, expressing the percentage of incidence of cSCC compared to all cancer diagnosis would be useful for the readers to understand the impact of the disease.

2.      Line 69: since authors report death rates of BCC and melanoma in addition to cSCC, it is important to explain that melanoma accounts for only 1% of all skin cancers, but due to its high aggressiveness it is responsible for the majority of skin cancer-related deaths (PMID: 34638427).

3.      In the paragraph 2.3 authors should at mention that there is an active research activity regarding the discovering on novel prognostic biomarkers, which aim to stratify patients better and provide more accurate prognosis (PMID: 35264873; PMID: 35264873; PMID: 35264873).

4.      A summary table of what reported in the paragraph 3.9 would improve the manuscript.

Reviewer 2 Report

The article is well-written and exhaustive. I would add a summary table of all treatments. Some paragraphs lack bibliographic citations.

More concise language would make the article easier to read.

Moreover, adjectives that express uncertainty reduce the impact of reading. Line 58 will likely further

Line 116 was roughly half

Line 154 it is generally

Line 18: change in an effort to with in efforts to

You should at least mention, in the treatment section, the possibility to use topical treatments as an adjuvant treatment; here two article you should add: doi: 10.3390/curroncol28040213. and doi: 10.3390/medicina57060563.

Round 2

Reviewer 2 Report

The paper improved after revisions. It is eligible to be published.